# Intravenous Iron Supplementation for the Treatment of Chemotherapy-Induced Anemia: A Systematic Review and Meta-Analysis of Randomized Controlled Trials

**DOI:** 10.3390/jcm11144156

**Published:** 2022-07-18

**Authors:** Shira Buchrits, Oranit Itzhaki, Tomer Avni, Pia Raanani, Anat Gafter-Gvili

**Affiliations:** 1Internal Medicine Department A, Rabin Medical Center, Beilinson Hospital, Petah-Tikva 4941492, Israel; ioranit@gmail.com (O.I.); boskko2001@gmail.com (T.A.); gaftera@gmail.com (A.G.-G.); 2Sackler Faculty of Medicine, Tel Aviv University, Tel Aviv 6997801, Israel; praanani@012.net.il; 3Institute of Hematology, Davidoff Cancer Center, Petah-Tikva 4941492, Israel

**Keywords:** chemotherapy-induced anemia, intravenous iron, functional iron deficiency

## Abstract

Background: The pathophysiology of cancer-related anemia is multifactorial, including that of chemotherapy-induced anemia (CIA). The guidelines are not consistent in their approach to the use of intravenous (IV) iron in patients with cancer as part of the clinical practice. Materials and methods: All randomized controlled trials that compared IV iron with either no iron or iron taken orally for the treatment of CIA were included. We excluded trials if erythropoiesis-stimulating agents (ESAs) were used. The primary outcome was the percentage of patients requiring a red blood cell (RBC) transfusion during the study period. The secondary outcomes included the hematopoietic response (an increase in the Hb level by more than 1 g/dL or an increase above 11 g/dL), the iron parameters and adverse events. For the dichotomous data, risk ratios (RRs) with 95% confidence intervals (Cis) were estimated and pooled. For the continuous data, the mean differences were calculated. A fixed effect model was used, except in the event of significant heterogeneity between the trials (*p* < 0.10; I^2^ > 40%), in which we used a random effects model. Results: A total of 8 trials published between January 1990 and July 2021 that randomized 1015 patients fulfilled the inclusion criteria. Of these, 553 patients were randomized to IV iron and were compared with 271 patients randomized to oral iron and 191 to no iron. IV iron decreased the percentage of patients requiring a blood transfusion compared with oral iron (RR 0.72; 95% CI 0.55–0.95) with a number needed to treat of 20 (95% CI 11–100). IV iron increased the hematopoietic response (RR 1.23; 95% CI 1.01–1.5). There was no difference with respect to the risk of adverse events (RR 0.97; 95% CI 0.88–1.07; 8 trials) or severe adverse events (RR 1.09; 95% CI 0.76–1.57; 8 trials). Conclusions: IV iron resulted in a decrease in the need for RBC transfusions, with no difference in adverse events in patients with CIA. IV iron for the treatment of CIA should be considered in clinical practice.

## 1. Background

Anemia is a common complication across all malignancies. According to a large European survey of 15,367 cancer patients, cancer-associated anemia has a prevalence of 39.3% at presentation and 67% within six months [1]. The pathophysiology of cancer-related anemia, including that of chemotherapy-induced anemia (CIA), is multifactorial and can comprise bleeding, an iron deficiency, an erythropoietin deficiency due to renal disease and tumor involvement of the bone marrow [2].

There is growing evidence that anemia has a negative impact in cancer. Anemia diminishes the functional capacity and is associated with a decrease in the performance status as well as the quality of life [3]. In addition, anemia increases the need for blood transfusions, which have been associated with transfusion reactions and infections [4].

One of the causes of CIA is functional iron deficiency (FID), defined as a defect in supplying iron to the erythroid marrow despite sufficient iron stores. Transferrin saturation (TSAT) is an indicator of iron availability for erythropoiesis. A low TSAT (<20%) and high ferritin (>100 ng/mL) suggest FID [5].

Erythroid-stimulating agents (ESAs) represent a therapeutic option for the treatment of CIA, but only 40–70% of patients with cancer obtain a hematological response. Several pre-clinical trials have identified potential safety problems related to ESAs [6,7,8,9,10,11]. One of the causes of the absence of an ESA response is FID [12]. To avoid FID, it has been suggested that ESAs should be administered with iron support [13,14]. 

Several pre-clinical trials have identified potential safety problems related to ESA exposure, suggesting that ESAs have a role in augmenting tumorigenesis and metastasis as well as increasing the risk of a thrombosis [9,10,11].

The current American Society of Clinical Oncology (ASH)/American Society of Hematology (ASH) guidelines do not recommend the adjunctive use of ESAs with chemotherapy when chemotherapy is administered with a curative intent. However, ESAs are considered by the guidelines for chemotherapy with a palliative intent [15]. This is based on evidence from two meta-analyses that demonstrated both increased mortality and venous thromboembolic events [16,17].

Oral iron is rarely used nowadays due to low tolerability and efficacy in patients, especially those with FID; no advantage was observed with oral iron when it was added to ESAs [18,19,20]. IV iron has previously been shown to increase the hematopoietic response and to reduce the need for RBC transfusions with no difference in mortality or adverse events in a meta-analysis of RCTs that assessed IV iron as an adjunct therapy with ESAs [21]. 

Given the fact that there is no consistent approach for IV iron use in cancer patients undergoing chemotherapy in clinical practice, we attempted to assess the effect of IV iron as a monotherapy for the treatment of chemotherapy-induced anemia.

## 2. Materials and Methods

### 2.1. Data Sources

We searched PubMed (January 1966 to July 2021), the Cochrane Central Register of Controlled Trials (CENTRAL) (The Cochrane Library, Issue 8, 12 August 2021) and the following conference proceedings for trials in oncology and hematology (2017–2021): Annual Meeting of the American Society of Hematology (ASH); Annual Meeting of the European Haematology Association (EHA); the American Society of Clinical Oncology (ASCO); and the European Society for Medical Oncology (ESMO). In addition, we searched databases of ongoing and unpublished trials: http://www.controlled-trials.com, http://www.clinicaltrials.gov/ct and http://clinicaltrials.nci.nih.gov. 

PubMed was searched using the terms: (iron OR sodium ferric gluconate OR iron dextran OR iron [MeSH] OR Iron-Dextran Complex [MeSH] OR ferric citrate OR Ferric Compounds [MeSH] OR oral* iron OR intravenous iron OR iv iron OR iron-gluconate OR ferrlecit OR iron-gluconate OR venofer OR ferrous sulphate) AND (cancer [MeSH] OR chemotherapy or malignancy or tumor) AND (Anemia or anemia [Mesh]) AND (randomized controlled trial [pt] OR controlled clinical trial [pt] OR randomized controlled trials [mh] OR random allocation [mh] OR double-blind method [mh] OR single-blind method [mh] OR ((singl* [tw] OR doubl* [tw] OR trebl* [tw] OR tripl* [tw]) AND (mask* [tw] OR blind* [tw])) OR (placebos [mh] OR placebo* [tw] OR random* [tw]) NOT (animals [mh] NOT human [mh]). 

We searched the relevant conferences using the term “iron”. The references quoted in all of the included trials and reviews were analyzed in order to identify any additional trials eligible for inclusion.

### 2.2. Study Selection 

We included all randomized controlled trials comparing IV iron with no iron or oral iron for treating anemia in cancer patients undergoing chemotherapy. All types of malignancies were included. Every IV iron preparation was included. Trials were included independently of the publication status, release date and language. Trials were excluded if any ESAs were used for any arm per protocol or off-label.

### 2.3. Data Extraction and Quality Assessment

Two reviewers independently extracted data from the included trials and evaluated the quality of the methodologies (S.B. and A.G-G). If the two reviewers were not in agreement, a third evaluator (O.I.) extracted the data and the results were obtained by a consensus. We assessed all possible sources of bias that were relevant, including allocation concealment, the generation of the allocation sequence, blinding, incomplete outcome data reporting and selective outcome reporting. We rated each domain as a low risk of bias, an unclear risk (lack of information on or uncertainty over the potential for bias) or a high risk of bias according to the criteria specified in the *Cochrane Handbook*, version 5.1.0. 

### 2.4. Definition of Outcomes

The primary outcome was the percentage of patients requiring an RBC transfusion during the study period. The secondary outcomes were divided into efficacy and safety outcomes. The efficacy outcomes included: the percentage of patients achieving a hematopoietic response defined as an increase in the hemoglobin (Hb) level by more than 1 g/dL or an increase above 11 g/dL; an absolute Hb concentration or a change from the baseline in the Hb concentration at the end of trial; the absolute ferritin level and transferrin saturation (TSAT) level at the end of the trial; or a change in these values from the baseline if the absolute values were unavailable. 

The safety outcomes included: any adverse event; severe adverse events that were considered serious according to the trial investigators or grade 3–5 adverse events according to the Common Terminology Criteria for Adverse Events (CTCAE version 4.03, NCI, Bethesda, MD, USA); gastrointestinal adverse events; or infusion reactions [22].

### 2.5. Data Synthesis and Analysis 

Our primary analysis was IV iron versus a control (no iron or oral iron). Dichotomous data were analyzed by calculating the risk ratio (RR) for each trial with a 95% confidence interval (CI) (Review Manager (RevMan), version 5.4 for Windows, The Cochrane Collaboration). For the continuous variables, we obtained the mean and standard deviation (SD). When the mean or SD values were not available, we calculated them by using data obtained from figures or by recalculating them from other effect estimates and dispersion measures. We calculated the mean difference (MD), which represented the combination of absolute differences between the mean values in the two groups in a clinical trial. This summary statistic had the same unit of measurement as the variable measured. Absolute end values rather than a change from the baseline values were preferentially analyzed. Where unavailable, we combined the end values and changes from the baseline values. 

We assessed the heterogeneity of the trial results by calculating a χ^2^ test of heterogeneity and the I^2^ measure of inconsistency. We used a fixed effect model with the Mantel–Haenszel method for pooling the trial results throughout the review unless a statistically significant heterogeneity was found (*p* = 0.10 or I^2^ > 50%), in which case we chose a random effects model and used the DerSimonian and Laird method [23]. We explored the potential sources of heterogeneity through subgroup analyses of the primary outcome according to the type of malignancy and the type of iron formulation. 

## 3. Results

### 3.1. Description of Included Studies

The search yielded 1683 potentially relevant publications, of which 42 were considered for a future investigation. In addition, one abstract from conference proceedings was included. The study flow chart according to the Preferred Reporting Items for Systematic Reviews and Meta-Analyses (PRISMA) guidelines showing the flow of the trials included in the meta-analysis and the reasons for exclusions is shown in Figure 1. After applying the inclusion criteria, 8 trials performed between January 1990 and July 2021 that randomized 1015 patients were selected [24,25,26,27,28,29,30,31]. Pooled together, 553 patients treated with IV iron were compared with 271 patients treated with oral iron and 191 treated with no iron.

Table 1 presents the characteristics of the included trials. Most trials included patients with solid tumors; three trials included patients with gynecologic cancers [24,25,26], one included esophageal cancer [30], one included lymphoproliferative malignancies [27] and three included all types of cancer [28,29,31].

All trials included patients with anemia. Each trial applied different inclusion criteria regarding the definitions for anemia and FID (Table 1). Four trials included patients with various degrees of anemia and FID, two trials included patients with anemia according gender (Hb < 12 g/dL for women and Hb < 13 g/dL for men) and two trials included patients with Hb < 10 g/dL. The ferritin level at the baseline was reported in three trials and ranged between 100 and 300 mg/dL [27,28,30]. The following IV iron formulations were used: iron sucrose in four trials, ferric carboxymaltose in two trials and IV iron isomaltoside (currently known as ferric derisomaltose) in two trials. In five trials, a fixed dose of IV iron was given; in three trials, the dose was calculated according to the hemoglobin and weight. The total dose of intravenously administered iron in the trials varied from 400 to 8000 mg. The IV iron schedule varied between the trials. All trials administered the iron after chemotherapy once a week; two trials administered a single dose, three trials administered two doses, two trials administered six doses and one trial administered eight doses. The follow-up ranged from 4 to 24 weeks. 

The risk of bias assessment is detailed in Table 2. Regarding the sequence generation, 75% of the trials were found to have a low risk of bias. Regarding the allocation concealment, 87.5% of the trials were of an unclear risk. All included studies were unblinded. All trials were considered to have a low risk of incomplete outcome data.

### 3.2. Primary Outcome: Transfusion Requirements

All eight trials reported the number of patients requiring an RBC transfusion following iron replacement. IV iron decreased the percentage of patients requiring an RBC transfusion compared with oral iron (RR 0.72; 95% CI 0.55–0.95; I^2^ = 23%), with a number needed to treat of 20 (95% CI 11–100) (Figure 2). The sensitivity analysis was restricted to studies that reported a low risk of sequence generation (*n* = 6) and did not alter the results (RR 0.78; 95% CI 0.58–1.06; I^2^ = 22%).

Subgroup analyses were performed according to the type of IV iron and the type of malignancy (Table 3). In the subgroup of gynecologic malignancies, the use of IV iron was associated with a decrease in the percentage of patients requiring an RBC transfusion compared with oral iron (RR 0.51; 95% CI 0.36–0.73; I^2^ = 0%; 5 trials). When analyzed according to the type of IV iron preparation, there was a significant decrease in the percentage of patients requiring an RBC transfusion in the four trials of iron sucrose (RR 0.67; 95% CI 0.47–0.94; I^2^ = 17%), but not in the two trials of ferric carboxymaltose or in the two trials of iron isomaltose.

### 3.3. Secondary Outcomes

IV iron increased the percentage of patients with a hematopoietic response compared with the control (RR 1.23; 95% CI 1.01–1.5; I^2^ = 0%).

The absolute Hb level or the change from the baseline in Hb at the end of the study was higher in patients treated with IV iron (MD 0.23; 95% CI 0.01–0.44).

Three trials (including 396 patients) reported on the parameters of iron indices. The ferritin level at the end of the trial was significantly higher in the IV iron arm compared with the standard treatment (MD 260.65; 95% CI 105.79–415.51). There was no difference in TSAT at the end of the trial between the patients treated with IV iron and those without (MD −0.4; 95% CI −5.96–5.17).

### 3.4. Safety

There was no difference between the study groups in the risk of any adverse events (RR 0.97; 95% CI 0.88–1.07; 8 trials) or severe adverse events (RR 0.98; 95% CI 0.47–2.06; I^2^ = 69%; random effects model; 8 trials) (Figure 3). There was no difference between the study groups in gastrointestinal adverse events (RR 1.05; 95% CI 0.85–1.3; 4 trials; I^2^ = 75%). There was no difference in the rate of adverse events requiring treatment discontinuation (RR 0.33; 95% CI 0.07–1.58; I^2^ = 81.2%; random effects model; 4 trials).

## 4. Discussion

In this systematic review and meta-analysis, we included all randomized controlled trials that compared IV iron with no iron or oral iron for chemotherapy-induced anemia (CIA). We found that the intravenous administration of iron for CIA reduced the risk of an RBC transfusion by 28% (95% CI 0.55–0.95; I^2^ = 23%). In addition, IV iron increased the chance of a hematopoietic response and was associated with an increase in the ferritin level and was not associated with an increase in adverse events (both any and severe). 

Our main finding of a decrease in the need for RBC transfusions is of importance to clinical practice. Lowering transfusion requirements can minimize various risks such as a hemolytic transfusion reaction, an acute lung injury and a transfusion infection-related acute lung injury [32,33]. It is a matter of debate if blood transfusions, especially if given peri-operatively, negatively impact cancer outcomes. In a large cohort study of more than 4000 patients with colorectal carcinomas undergoing curative colorectal resections, blood transfusions administered peri-operatively were found to be independently associated with shorter disease-free survival as well as overall survival [34]. This was independent of the anemia status. Although the setting in this study was different from patients undergoing chemotherapy (as in our study), these results further reinforce the need for restrictive transfusion strategies. IV iron, as we have shown, has the advantage of minimizing transfusions. 

Our results are in accordance with the current guidelines for blood transfusions, which recommend a restrictive transfusion strategy [35]. The need for fewer RBC transfusions may also potentially reduce hospital visits (either to the hospital ambulatory day clinic or hospitalization), which may have a positive impact on the quality of life [36].

The increase in hematopoietic response is of a clinical relevance. Anemia has been shown to be a negative prognostic factor in cancer [37,38].

IV iron was not associated with an increased risk of adverse events. Similar safety results were shown in a comprehensive meta-analysis that included 103 randomized trials and 14,434 patients in which IV iron was compared with oral iron in many different clinical settings. IV iron was shown to have a comparable safety profile to oral iron with the same risk of SAEs, mortality and serious bacterial infections and fewer GI adverse events [39].

Previous clinical trials using IV iron and a previous meta-analysis support the use of IV iron in CIA. IV iron has been shown to improve the hematopoietic response, to reduce the risk of RBC transfusions and to be well-tolerated [21]. However, this was shown in trials that administered both IV iron and ESAs. This current study is the first meta-analysis to assess the benefit of IV iron supplementation as a monotherapy without ESAs in CIA. As mentioned above, ESAs are controversial because of potential safety problems and are currently only recommended for palliative care [9,10,11].

Our results are in line with most consensus guidelines that recommend IV iron supplementation for the treatment of CIA. The European Society for Medical Oncology (ESMO) originally suggested a treatment with IV iron in their 2010 guidelines and confirmed the utility of IV iron in their 2018 update [40]. The EORTC (European Organisation for Research and Treatment of Cancer) guidelines mention a better response to ESAs with iron IV, but indicate the need to define the optimum dose and timing [9].

Our study has several limitations that merit consideration; first, the included studies were heterogeneous with respect to the various types of malignant tumors and chemotherapy regimens. In addition, there was heterogeneity regarding the iron supplementation, including different iron preparations and schedules. There was not enough information to conduct subgroup analyses according to different baseline hematologic parameters or different malignancies or according to the total administered iron dose. There was only one trial that assessed hematological malignancies. The optimal iron dosage and schedule was not clear. Due to the short follow-up period of up to 24 weeks, there was no long-term follow-up data regarding efficacy, mortality and safety. In addition, we could not collect data regarding the effect of IV iron on cancer-related outcomes. Data regarding the cost-effectiveness of IV iron were not collected as well.

### Implications for Practice and Research

Our meta-analysis supports the use of iron intravenously administered for the treatment of CIA. Our results mainly apply to patients with FID. Further research is needed to define the optimal IV iron formulation, dose and schedule and to assess the specific types of malignancies that may benefit from IV iron.

In conclusion, in this meta-analysis we showed that IV iron for the treatment of CIA reduces the need for RBC transfusions and is not associated with adverse events. IV iron for the treatment of CIA should be considered in clinical practice.

## Figures and Tables

**Figure 1 jcm-11-04156-f001:**
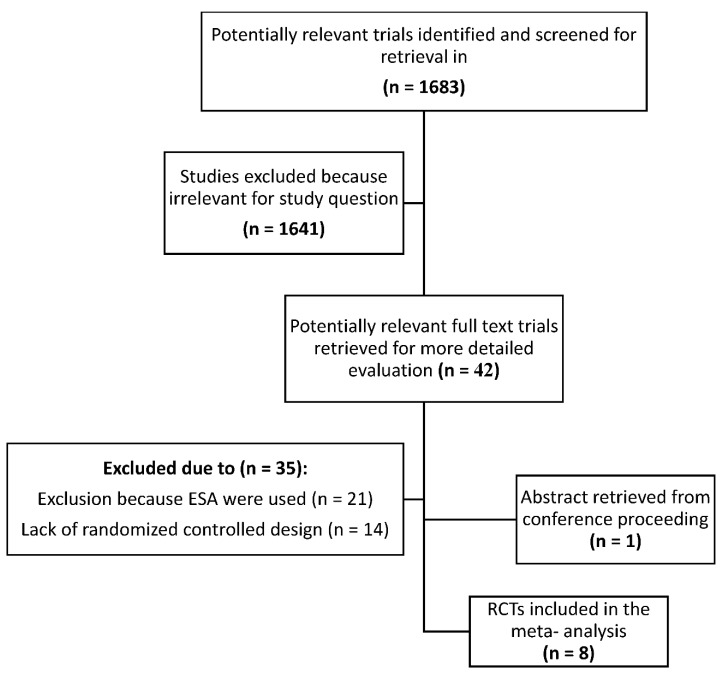
Trial flow according to Preferred Reporting Items for Systematic Reviews and Meta-Analyses (PRISMA) guidelines showing the flow of trials included in the meta-analysis. ESA: erythropoiesis-stimulating agents; RCT: randomized controlled trial.

**Figure 2 jcm-11-04156-f002:**
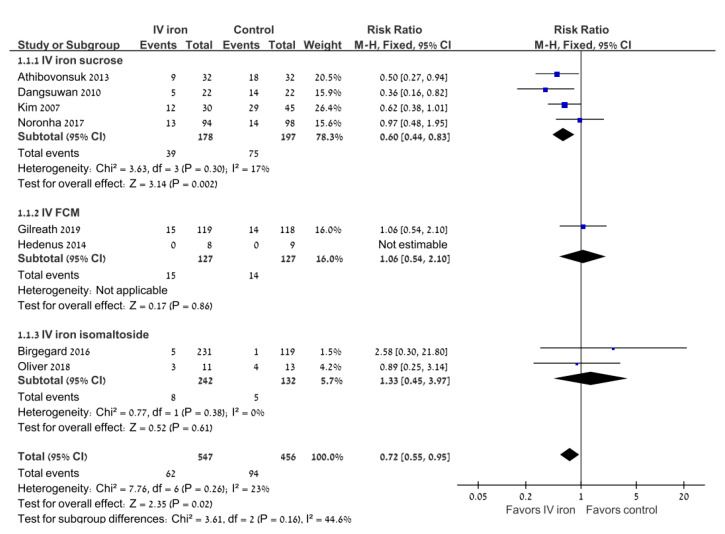
Intravenous iron versus the control: the need for RBC transfusions. Blue squares represent the point estimate, their sizes represent their weight in the pooled analysis and the horizontal bars represent the 95% CI. The black diamond at the bottom represents the pooled point estimate. CI: confidence interval; IV: intravenous.

**Figure 3 jcm-11-04156-f003:**
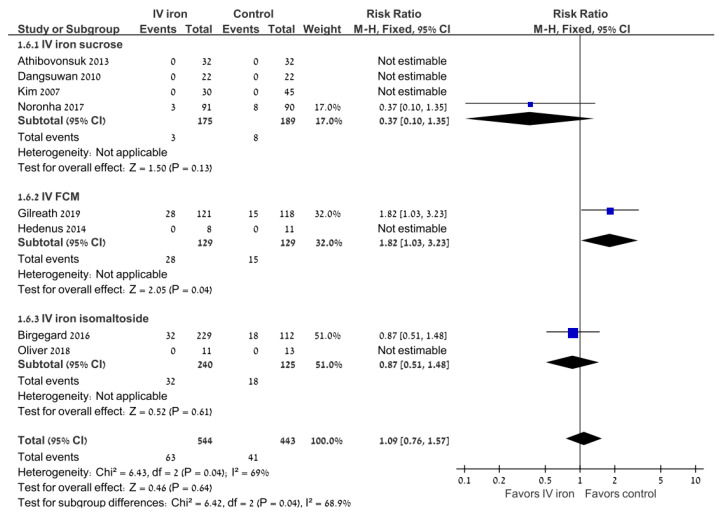
Intravenous iron versus the control: severe adverse events. Blue squares represent the point estimate, their sizes represent their weight in the pooled analysis and the horizontal bars represent the 95% CI. The black diamond at the bottom represents the pooled point estimate. CI: confidence interval; IV: intravenous.

**Table 1 jcm-11-04156-t001:** Characteristics of trials.

Study	Treatment Arms	Number of Randomized Patients	IV Iron Type and Dosing Schedule	Type of Malignancy	Inclusion Criteria	Hb and Ferritin at Baseline	TSAT at Baseline	Ferritin at Baseline
Kim 2007 [24]	IV iron sucrose	30	Iron sucrose 200 mg, one single dose after chemotherapy infusion for a maximum of 6 weeks	All patients with cervical cancer treated with concurrent chemoradiotherapy	Hgb < 12 g/dL and Hgb > 10 g/dL	11.27 ± 1.94 g/dL	Not reported	Not reported
	No iron	45				11.33 ± 2.14 g/dL		
Dangsuwan 2010 [25]	IV iron sucrose	22	Iron sucrose 200 mg, one single dose after chemotherapy infusion	All patients with ovarian or endometrial cancer receiving platinum-based chemotherapy	Hgb < 10 g/dL	8.9 ±0.6 g/dL	Not reported	Not reported
	Oral iron	22	Ferrous fumarate 600 mg daily			9 ±0.6 g/dL		
Athibovonsuk 2013 [26]	IV iron sucrose	32	Iron sucrose 200 mg, one single dose after chemotherapy infusion for 6 weeks	All patients with ovarian or endometrial cancer receiving platinum-based chemotherapy	Hgb < 12 g/dL (women) or Hgb < 13 g/dL (men)	11.3 ± 0.8 g/dL	Not reported	Not reported
	Oral iron	32	Ferrous fumarate 600 mg daily during the treatment period			11.4 ± 1 g/dL		
Hedenus 2014 [27]	IV iron carboxymaltose	8	Ferric carboxymaltose 1000 mg weekly for 8 weeks	All patients with lymphoid malignancies	Hgb < 10.5 g/dL and FID TSAT < 20%, ferritin > 30 ng/mL (women) or > 40 ng/mL (men)	Hb 9.5 (9–10.5) g/dL Ferritin 216 (65–800) ng/mL	16 (3–35)	216 (65–800) mcg/L
	No iron	11				Hb 9.8 (8.4–10.6) g/dL Ferritin 322 (8–707) ng/mL	18 (0–31)	322 (8–707) mcg/L
Birgegård 2016 [28]	IV iron isomaltoside (infusion)	114	Iron isomaltoside up to 1000 mg for a maximum of 2 weeks	All types	Hgb < 12.0 g/dL, TSAT < 50%, serum ferritin < 800 ng/ml	Hb 10.6 ± 8.7 g/dL Ferritin 254.2 ± 290.3 ng/mL	58.1 ± 13.5	254.2 ± 290.3 mcg/L
	IV iron isomaltoside (bolus)	117	Iron isomaltoside 500 mg weekly for 4 weeks			Hb 14.1 ± 14.4 g/dL Ferritin 222.0 ± 207.9 ng/mL	60.1 ± 14.6	222.0 ± 207.9 mcg/L
	Oral iron	119	Iron sulfate 200 mg daily for 12 weeks			Hb 14.5 ±11.9 g/dL Ferritin 247.4 ± 254 ng/mL	58.9 ± 13.3	247.4 ± 254 mcg/L
Noronha 2017 [29]	IV iron sucrose	94	Iron sucrose 760 mg after chemotherapy infusion for 2 weeks	All types	Hgb < 10 g/dL and TSAT < 20%	10.2 (7,2–11.9) g/dL	Not reported	Not reported
	Oral iron	98	300 mg daily for 2 weeks			10.1 (7.2–12.5) g/dL		
Oliver 2018 [30]	IV iron isomaltoside	14	Iron isomaltoside, single dose	All patients with esophageal adenocarcinoma	Hgb < 12 g/dL (women) or Hb < 13 g/dL (men)	Hb 9.96 g/dL Ferritin 105 (120) ng/mL	Not reported	Not reported
	No iron	13				Hb 11.45 g/dL Ferritin 161 (123) ng/mL		
Jeffrey A. Gilreath 2019 [31]	IV iron carboxymaltose	122	15 mg/kg (750 mg max) for 2 weeks	All types	Hgb < 11 g/dL and FID, TSAT < 35%, ferritin 100–800 ng/mL	Not reported	Not reported	Not reported
	No iron	122						

FID: functional iron deficiency; Hgb: hemoglobin; IV: intravenous; TSAT: transferrin saturation.

**Table 2 jcm-11-04156-t002:** Risk of bias assessment.

Study	Sequence Generation	Allocation Concealment	Blinding	Incomplete Outcome Data	Selective Outcome Reporting
Kim 2007 [24]	Low risk	Unclear risk	None	Low risk	Low risk
Dangsuwan 2010 [25]	Unclear risk	Unclear risk	None	Low risk	Low risk
Athibovonsuk 2013 [26]	Low risk	Unclear risk	None	Low risk	Low risk
Hedenus 2014 [27]	Low risk	Unclear risk	None	Low risk	Low risk
Birgegård 2016 [28]	Low risk	Unclear risk	None	Low risk	Low risk
Noronha 2017 [29]	Low risk	Unclear risk	None	Low risk	Low risk
Oliver 2018 [30]	Unclear risk	Unclear risk	Only patient-blinded	Low risk	Low risk
Jeffrey A. Gilreath 2019 [31]	Low risk	Low risk	Double-blind	Low risk	Low risk

**Table 3 jcm-11-04156-t003:** Subgroup analyses of the primary outcome: the need for Red Blood Cell (RBC) transfusions.

		Relative Risk	95% Confidence Interval	Number of Trials
Analysis according to type of IV iron preparation	Iron sucrose	0.67	0.47–0.94	4
	Iron ferric carboxymaltose	1.06	0.54–2.1	2
	Iron isomaltoside	2.58	0.3–21.8	2
Analysis according to type of malignancy	Solid tumors	1.06	0.68–1.66	5
	Lymphoproliferative malignancy			1
	Gynecologic malignancy	0.51	0.36–0.73	3

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
