# Peer review of "Intravenous Iron Supplementation for the Treatment of Chemotherapy-Induced Anemia: A Systematic Review and Meta-Analysis of Randomized Controlled Trials"

_jcm, 2022, doi:10.3390/jcm11144156_

Round 1

Reviewer 1 Report

I do not have major concerns about the data presented in the paper. Authors may consider discussing in more details following points:

  • Is it not clear if a lower transfusion requirement relates or not to response to treatment and overall survival of the patients and if IV iron relates with patients outcome. Are there any data about?
  • A doubt about the sentence “The need for less transfusions may also potentially reduce visits to the ambulatory day 241 clinic, which may have a positive impact on quality of life. [35]”. Do the patients need accesses to the hospital to undergo IV iron therapy?
  • It would be also advisable to speculate about direct and indirect costs to be sustained.

There are also few grammatic/typos mistakes that I would suggest to correct (e. g. Line 73 on page 5 – “Oral iron has is rarely used nowadays”.

Overall, I am supportive for this paper to be reconsidered.

Author Response

We thank the editor and the reviewers for the chance to revise our manuscript. We had answered the reviewers’ remarks, point by point and revised accordingly. Among the changes in this revised manuscript, we removed misprints and corrected copy errors. We believe that the manuscript following the revision is now clearer, and the results are better explained.

Sincerely,

Dr. S. Buchrits

Reviewer 1 report:

I do not have major concerns about the data presented in the paper. Authors may consider discussing in more details following points:

  • Is it not clear if a lower transfusion requirement relates or not to response to treatment and overall survival of the patients and if IV iron relates with patients outcome. Are there any data about?

We show in this study that IV iron administered to cancer patients undergoing therapy reduces transfusion requirements. The trials included were trials that only assessed hematologic response (increase in Hb), iron parameters, transfusion requirements and quality of life. There are no data in the original trials regarding response to chemotherapy or long-term outcomes as survival. Therefore we could not collect long term survival data and accordingly we could not show any correlation between lower transfusion requirements and  response or long term outcomes. We added this to the limitations section.

  • A doubt about the sentence “The need for less transfusions may also potentially reduce visits to the ambulatory day 241 clinic, which may have a positive impact on quality of life. [35]”. Do the patients need accesses to the hospital to undergo IV iron therapy?

Patients must have access to a hospital in order to receive blood transfusion. However, IV iron may be given in ambulatory clinics. We rephrased the relevant sentence and now it is accurate.  

  • It would be also advisable to speculate about direct and indirect costs to be sustained.

We agree that cost effective analysis may be very interesting. However, data were not reported in the original trials, thus could not be pooled. We show that IV iron reduces transfusions. Since there is no formal cost effectiveness analysis we can only speculate that administration of IV iron may reduce the costs of RBC transfusions, hospital visits, hospitalization etc

  • There are also few grammatic/typos mistakes that I would suggest to correct (e. g. Line 73 on page 5 – “Oral iron has is rarely used nowadays”.

corrected

Overall, I am supportive for this paper to be reconsidered.

Reviewer 2 Report

I have received for reviewing an interesting paper entitled “Intravenous iron supplementation for the treatment of chemotherapy-induced anemia – systematic review and meta-analysis of randomized controlled trials”. Here are my comments:

- line 34 – “tooral”

- lines 41-43 – please to rewrite in order to avoid repeating words

- lines 51, 55 and so on – please don’t place period before reference

- lines 52-55 – what about the relation between blood transfusion and cancer recurrence? It is one of the biggest issues and there are various studies about this (e.g. Wu, HL., Tai, YH., Lin, SP. et al. The Impact of Blood Transfusion on Recurrence and Mortality Following Colorectal Cancer Resection: A Propensity Score Analysis of 4,030 Patients. Sci Rep 8, 13345 (2018). https://doi.org/10.1038/s41598-018-31662-5)

- lines 62 and 65 – please try to rewrite

- line 169 - what do you mean by “treated with no iron”? What was the treatment, excluding the ESA? No treatment at all?

- lines 236-242 – as I emphasized before, the link between blood transfusion and cancer recurrence should be mentioned

- lines 240-242 – but patients should have visits to the ambulatory day clinic to receive IV iron. From the eight presented trials, 6 of them reported at least two administered doses “after chemotherapy once a week” – maybe this is not a good angle

- 243-244 – please present some of those factors associated with the “negative prognostic in cancer”

- line 259-261 – please do not repeat the exact information

- line 278 – “Debate still exists regarding Further research is”

- I think that in Discussion section you should speak about FID in cancer and chemotherapy, and FID and IV iron administration – this should be a systemic review and all aspects should be covered

- according to Journal recommendation, references should be described as follows: Journal Articles: 1. Author 1, A.B.; Author 2, C.D. Title of the article. Abbreviated Journal Name Year, Volume, page range.

Author Response

Reviewer 2 report:

I have received for reviewing an interesting paper entitled “Intravenous iron supplementation for the treatment of chemotherapy-induced anemia – systematic review and meta-analysis of randomized controlled trials”. Here are my comments:

- line 34 – “tooral”

corrected

- lines 41-43 – please to rewrite in order to avoid repeating words

Corrected

- lines 51, 55 and so on – please don’t place period before reference

corrected

- lines 52-55 – what about the relation between blood transfusion and cancer recurrence? It is one of the biggest issues and there are various studies about this (e.g. Wu, HL., Tai, YH., Lin, SP. et al. The Impact of Blood Transfusion on Recurrence and Mortality Following Colorectal Cancer Resection: A Propensity Score Analysis of 4,030 Patients. Sci Rep 8, 13345 (2018). https://doi.org/10.1038/s41598-018-31662-5)

We show in this study that IV iron administered to cancer patients undergoing therapy reduces transfusion requirements. The trials included were trials that only assessed hematologic response (increase in Hb) , iron parameters, transfusion requirements and quality of life. There are no data in the original trials regarding response to chemotherapy or recurrence of cancer. We think that the reference the reviewer suggested is of great importance. As stated the effect of RBC transfusions on cancer recurrence is controversial. Since our intervention was IV iron (and not RBC transfusions) , the outcome of those who received RBC transfusions vs those who did not could not be evaluated. Moreover, there were no data in these trials regarding cancer recurrence.

We therefore added this issue in the discussion section to emphasize the importance of the decrease in RBC transfusions due to possible negative effects of transfusions on cancer recurrence. We quoted the Wu study as suggested.

- lines 62 and 65 – please try to rewrite

corrected

- line 169 - what do you mean by “treated with no iron”? What was the treatment, excluding the ESA? No treatment at all?

No iron at all

- lines 236-242 – as I emphasized before, the link between blood transfusion and cancer recurrence should be mentioned

We agree. We added this association to the discussion section. We could not add data to results since this was not reported in trials

- lines 240-242 – but patients should have visits to the ambulatory day clinic to receive IV iron. From the eight presented trials, 6 of them reported at least two administered doses “after chemotherapy once a week” – maybe this is not a good angle

- 243-244 – please present some of those factors associated with the “negative prognostic in cancer”

- line 259-261 – please do not repeat the exact information

Corrected

- line 278 – “Debate still exists regarding Further research is”

Corrected

Round 2

Reviewer 1 Report

the paper in the present form can be accepted

Author Response

thank you

Reviewer 2 Report

Congratulations to the authors for the improvements made to the manuscript!

To begin with, the initial comment about transfusion-associated cancer recurrence was an element in favor of the importance of this article …

Further, there are only a few minor issues that need to be solved, in my opinion.

Minor issues

- line 111 “analyzedin order…” – please correct

- line 234 – place a period at the end of the sentence

- line 284 – explain EORTC

- lines 285-286 – “but not orally…..but indicate….” – please rewrite

Author Response

corected

This manuscript is a resubmission of an earlier submission. The following is a list of the peer review reports and author responses from that submission.